# Factors Associated with Medication Non-Adherence among Patients with Lifestyle-Related Non-Communicable Diseases

**DOI:** 10.3390/pharmacy9020090

**Published:** 2021-04-22

**Authors:** Rie Nakajima, Fumiyuki Watanabe, Miwako Kamei

**Affiliations:** 1School of Pharmacy, Nihon University, Chiba 274-8555, Japan; watanabe.fumiyuki@nihon-u.ac.jp; 2Faculty of Pharmaceutical Sciences, Teikyo Heisei University, Tokyo 164-8530, Japan; m.kamei@thu.ac.jp

**Keywords:** unintentional non-adherence, intentional non-adherence, non-communicable diseases, health locus of control

## Abstract

This cross-sectional study explored the association between medication non-adherence and its factors in patients with non-communicable diseases (NCDs) using an online structured questionnaire emailed to 30,000 people (aged over 20 years who lived in Japan at the time of the survey). The questions concerned respondents’ characteristics, medication non-adherence, health beliefs, lifestyles, and trouble taking medication. Factors related to non-adherence were analyzed among patients with lifestyle-related NCDs categorized into two age groups: 20–59, and >60 years. Unintentional (*p* < 0.001) and intentional (*p* < 0.001) non-adherence were more common among patients aged 20–59 than in older adults. NCD patients aged 20–59 experienced significantly more trouble taking medication than older adults. Multiple regression analysis showed that for patients aged 20–59 with NCDs, unintentional non-adherence was significantly and positively associated with current smoking habits (β = 0.280, *p* < 0.001), while intentional non-adherence was significantly and positively associated with alcohol consumption (β = 0.147, *p* = 0.020) and current smoking habits (β = 0.172, *p* = 0.007). In patients aged 20–59, unhealthy eating habits (β = −0.136, *p* = 0.034) and lack of exercise (β = −0.151, *p* = 0.020) were negatively associated with intentional non-adherence. In conclusion, factors affecting medication non-adherence in patients with lifestyle-related diseases are related to health awareness, lifestyle, and medication barriers.

## 1. Introduction

The risk of morbidity of non-communicable diseases (NCDs), such as diabetes, high blood pressure, cerebrovascular disease, and cancer, increases owing to unhealthy lifestyle habits, including excessive alcohol consumption, smoking, and unhealthy eating habits [1]. Since these diseases require long-term treatment and lead to an increase in medical costs, many countries take measures to prevent NCDs [2,3]. Poor medication adherence by patients with NCDs affects disease prognoses; thus, it is essential to focus efforts on its improvement [4].

To improve medication adherence, it is necessary to identify the factors that constitute barriers to medication adherence among patients. Factors related to medication non-adherence have been identified by various studies, and the relationship between socioeconomic factors such as sex, age, race, and educational background and non-adherence is well known [5,6]. It has also been suggested that the personality and health beliefs of patients influence medication non-adherence [7]. The health locus of control (HLC) is one of the indicators used to examine the characteristics associated with patients’ beliefs about health, and it is related not only to non-adherence to medication but also to self-care and lifestyle-related NCDs [8,9]. Previous research has found a strong relationship between lifestyle habits such as smoking, drinking alcohol, and a lack of exercise and non-adherence to medication [10]. Furthermore, difficulties directly related to medication intake, such as concerns about side effects and the degree of illness or disability of patients have also been identified as factors influencing medication non-adherence [6,11].

Indicators of unintentional and intentional non-adherence can be measured by distinguishing whether the patient is not taking the drug on purpose or unintentionally. Such a distinction aids the development of specific and practical interventions aimed at improving adherence [12,13].

In recent years, an increase in the number of young people with NCDs has been reported as a result of changes in lifestyle habits [14,15]. Furthermore, young people with lifestyle-related NCDs often show poor medication adherence [16,17]. Younger patients with NCDs need longer and more consistent medication support to prevent a poor prognosis, as they are more likely to continue needing medication for longer periods of time [17]. Poor adherence among young people with lifestyle-related NCDs is expected to have a unique association with the factors influencing adherence in this age group.

In 2017, the numbers of patients with diabetes, dyslipidemia, and hypertension in Japan were 3.29 million, 2.21 million, and 9.94 million, respectively [18]. In particular, the morbidity and death rate of lifestyle-related diseases in the working generation (60 years old or younger) has become a problem. In 2019, for those in their 40 s, heart disease and cerebrovascular disease were the third and fourth leading causes of death, respectively; for those in their 50 s, heart disease was the second leading cause of death [19]. It is expected that patients’ beliefs about health and lifestyles differ from country to country; therefore, it is meaningful to conduct a survey in Japan to improve the evidence regarding medication adherence under various circumstances.

Therefore, this study aimed to identify the association between medication non-adherence and its factors in patients with NCDs by age group.

## 2. Materials and Methods

### 2.1. Study Design and Participants

A cross-sectional study was conducted anonymously using an online questionnaire. Data collection was conducted from 1–5 March 2019. An Internet research company (Nextit Research Institute, Inc., Kobe, Japan) randomly selected approximately 30,000 people (people over 20 years of age who lived in Japan at the time of the survey) from their registered users (approximately 150,000) and sent an email to all selected individuals simultaneously, inviting them to participate in a questionnaire survey. The inclusion criteria for the study were adults diagnosed with lifestyle-related NCDs (hypertension, diabetes, dyslipidemia, gout, heart disease, cerebrovascular disease, and cancer).

### 2.2. Sample Size

The Raosoft^®^ (Raosoft, Inc., Seattle, WA, USA) online sample size calculator was used to determine the sample size [20]. Based on this, a sample size of 385 was required for a confidence level of 95% and a 5% margin of error. The analysis only included those who responded to the request email and those who also suffered from lifestyle-related diseases. A response from 2000 people was required based on the 20 to 30% prevalence of lifestyle-related diseases according to a survey by the Ministry of Health, Labor and Welfare [21]. Furthermore, based on prior data from the research company, contacting 30,000 people was advised to obtain a response from more than 2000 people within the survey period (the survey for this study took place at the end of the fiscal year in Japan, the busiest period, and the response rate was expected to be low).

### 2.3. Measures

#### 2.3.1. Demographic Data

The questionnaire included questions regarding respondent age and sex, history of NCDs, and lifestyle habits (alcohol consumption more than three days a week, smoking status at the time of study, whether they felt they got sufficient sleep, whether they were concerned about their eating habits, and whether they had regular exercise habits).

#### 2.3.2. Health Locus of Control (HLC)

Patients’ beliefs about health were measured using the Japanese version of the Health Locus of Control (HLC) Scale [22]. HLC refers to people’s beliefs regarding what determines the status of their health. The HLC Scale consists of five subscales with five dimensions each (25 items in total): Internal, Family, Professional, Chance, and Supernatural. For example, people who have high HLC Internal believe that their behavior and efforts determine their health condition; in contrast, people who have high HLC Chance believe that their luck affects their health condition. Each item was rated on a six-point Likert-type scale ranging from 1 = *strongly disagree* to 6 = *strongly agree*. Scale items are listed in Table 1. Cronbach’s α coefficients for the HLC factors in this study were Supernatural = 0.892, Chance = 0.838, Internal = 0.839, Family = 0.867, and Professional = 0.835.

#### 2.3.3. Unintentional and Intentional Non-Adherence

In this study, we used a measure of patients’ unintentional and intentional non-adherence, which has been used before, and its reliability and validity have been established [6,23]. Unintentional non-adherence means that the patient unintentionally does not follow the suggestions of a professional, while intentional non-adherence is a deliberate failure to adhere to the suggestions of healthcare professionals. The subscale of unintentional non-adherence consists of four items and the subscale of intentional non-adherence consists of five items. Each item was rated on a five-point Likert-type scale ranging from 1 = *very often* to 5 = *never*. Total scores on the unintentional non-adherence subscale range from 4 to 20, with lower scores indicating higher levels of non-adherence. Scores on the intentional non-adherence subscale range from 5 to 25, with lower scores indicating higher levels of non-adherence. Scale items are listed in Table 2. The Cronbach’s α for unintentional non-adherence was 0.903 compared with 0.952 for intentional non-adherence.

#### 2.3.4. Trouble Taking Medication

The following seven questions were asked to assess respondents’ possible trouble taking medication. Question items were created based on the findings of previous research and clinical experience of pharmacists [24]. The question items were as follows: “I have experienced side effects”; “Sometimes the medicine is hard for me to swallow”; “Sometimes I can’t handle medicine tools” (medicine tools are devices for medicine use; e.g., insulin self-injection device, inhaler device for respiratory medicines); “Sometimes I can’t hear what a healthcare professional says”; “Sometimes I feel the explanations from doctors and pharmacists are difficult to understand”; “Sometimes I forget what my doctor or pharmacist told me”; and “I find it difficult to organize my medicines.” The participants could answer “yes” or “no” to the questions. The Kuder Richardson 20 of this index was α = 0.817.

### 2.4. Data Analysis

Patient characteristics were analyzed by age group using chi-square tests. Unintentional and intentional non-adherence by patient age group and HLC were compared using *t*-tests. In addition, differences in lifestyle and trouble taking medication by age were compared using chi-square tests. The relationship between patients’ HLC and unintentional and intentional non-adherence was analyzed using Pearson’s correlation. A multiple regression analysis was used to examine the influence of lifestyle and trouble taking medication on unintentional and intentional non-adherence by age. Patients were categorized into two age groups: 20–59 years and 60 years or older. In Japan, the average age of retirement is 60 years, and after retirement, individuals’ lifestyles and living environments change drastically, which was the reason these two categories were established in this study. The statistical software SPSS Statistics 26 (SPSS Inc., Chicago, IL, USA) was used for the analysis. The significance level was set to *p* < 0.05.

### 2.5. Ethical Considerations

The survey protocol adhered to the ethical guidelines for medical and health research involving human subjects and was approved by the ethical review board of Nihon University’s School of Pharmacy (approval number: 18-029, approval date: 12 February 2019). The survey was conducted with only those respondents who provided consent. Respondents were informed that there was no disadvantage in refusing to participate, and information concerning the protection of personal information was included at the beginning of the questionnaire.

## 3. Results

During the study period, a total of 2186 individuals responded to the questionnaire (response rate of approximately 7%). Of these, 599 (27.4%) who had a history of lifestyle-related NCDs were included in the analysis. The youngest participant was 20 years old and the eldest was 74 years old. Characteristics of patients with lifestyle-related NCDs are shown in Table 3.

### 3.1. Unintentional and Intentional Non-Adherence, HLC, Lifestyle Habits, and Trouble Taking Medication by Age Group

Both unintentional and intentional non-adherence were higher in patients aged 20–59 years than among patients aged over 60 years (unintentional *p* < 0.001, intentional *p* < 0.001). For patients aged 20–59 years, HLC was significantly lower for the Internal (*p* = 0.001) and Family (*p* = 0.012) subscales, while the Supernatural (*p* = 0.001) and Chance (*p* = 0.034) subscales were significantly higher among patients aged over 60 years. Young patients exhibited more unhealthy lifestyle habits, such as current smoking (*p* = 0.001), lack of sleep (*p* < 0.001), and indifference to eating habits (*p* < 0.001) than patients aged over 60 years. Patients aged 20–59 years reported significantly more trouble taking medication than older adults, for all items (Table 4).

### 3.2. Correlations between Unintentional and Intentional Non-Adherence and Types of HLC by Age

An analysis of the relationship between unintentional and intentional non-adherence and types of HLC using Pearson’s correlation revealed a positive association between the Internal subscale and unintentional and intentional non-adherence in patients aged 20–59 years (unintentional *r* = 0.274, *p* < 0.01; intentional *r* = 0.396, *p* < 0.01). Furthermore, a positive correlation was found between intentional non-adherence and the Family and Professional subscales of HLC (F: *r* = 0.196, *p* < 0.05; Pr: *r* = 0.160, *p* < 0.05). The Internal subscale was also positively correlated with unintentional and intentional non-adherence (unintentional *r* = 0.170, *p* < 0.01, intentional *r* = 0.232, *p* < 0.01) among patients aged over 60 years. In this age group, the Supernatural (unintentional r = −0106, *p* < 0.05, intentional r = −0.222, *p* < 0.01) and Chance (unintentional r = −0.164, *p* < 0.01, intentional r = −0.172, *p* < 0.01) subscales of HLC were negatively correlated with unintentional and intentional non-adherence (Table 5).

### 3.3. Unintentional and Intentional Non-Adherence, Lifestyle Habits, and Trouble Taking Medication by Age

Table 6 shows the results of a multiple regression analysis of factors related to medication non-adherence, lifestyle habits, and trouble taking medication. Among the lifestyle habits of patients aged 20–59 years, current smoking (β = 0.280, *p* < 0.001) was significantly and positively associated with unintentional non-adherence. For this age category, consuming alcohol more than three days a week (β = 0.147, *p* = 0.020) and current smoking habits (β = 0.172, *p* = 0.007) were significantly and positively associated with intentional non-adherence. Furthermore, unhealthy eating habits (β = −0.136, *p* = 0.034) and a lack of exercise (β = −0.151, *p* = 0.020) were significantly and negatively associated with intentional non-adherence. Sleep deprivation (β = 0.164, *p* = 0.002) was significantly and positively associated with unintentional non-adherence among patients aged over 60 years. In patients aged 20–59 years, there were positive associations of unintentional and intentional non-adherence with some trouble taking medicine: “Sometimes I cannot hear what a healthcare professional says” (unintentional β = 0.245, *p* = 0.004; intentional β = 0.336, *p* < 0.001) and “I find it difficult to organize my medicines” (unintentional β = 0.132, *p* = 0.039; intentional β = 0.164, *p* = 0.014).

## 4. Discussion

Both intentional and unintentional medication non-adherence among patients with lifestyle-related NCDs were found to be higher in the younger generation. Many previous studies have demonstrated that younger generations show poor medication adherence, which is consistent with the results of the present research [12,25].

Regarding health beliefs, patients aged 20–59 years with lifestyle-related diseases tended to have low internal HLC. High internal HLC showed a positive correlation with lower unintentional and intentional non-adherence, which suggests that healthcare services should actively strive to raise patients’ internal HLC. This study also showed that a high score on the Professional subscale of HLC was associated with better intentional adherence among 20–59 year-old patients. In general, it has been shown that better medication adherence is seen among patients with high Professional HLC, and an interaction between the Internal and Professional subscales of HLC is expected to lead to further improvement in medication adherence [26]. Our study also revealed that high Supernatural and Chance HLC influence non-adherence. Therefore, it is necessary to motivate adherence to medication through self-management and effort rather than through God or luck. Thus, healthcare professionals may need to work to improve patient health literacy and, when required, enlist the help of clergymen.

In terms of trouble taking medication in patients aged 20–59 years with lifestyle-related diseases, they tended not to hear the explanations of doctors and pharmacists, or forgot these explanations, which are issues affecting adherence. As a basis for communication between healthcare professionals and patients, repeated guidance and speaking in a clear and loud manner are recommended when communicating with patients over 60 years [27]. However, the present results show that similar consideration is also needed for younger patients with lifestyle-related diseases. It is important to follow these guidelines to address the dissatisfaction of patients aged 20–59 years, such as the inability to hear the healthcare professional’s explanations and increase their trust. Such an intervention is expected to be effective in terms of the prevention and reduction of intentional medication non-adherence. Effort on the part of pharmaceutical companies is necessary to develop easy-to-swallow medicine and easier-to-use devices, keeping in mind patients who struggle with swallowing medicine and using medicine devices.

Furthermore, other studies have revealed that people with lifestyle-related diseases at a younger age tend to have low socioeconomic status, such as a low educational background [28]. Therefore, having more trouble taking medication may be affected by the respondents’ low socioeconomic status. However, in this study, socioeconomic factors could not be pursued in depth because there were no data on the respondents’ income and educational history due to a lack of appropriate survey items.

Patients who follow a healthy diet often show good medication adherence [29]. However, this study found that patients aged 20–59 years with lifestyle-related diseases who did not care about their diet and/or exercise habits showed better results on intentional non-adherence. Previous studies have shown that patients with diabetes believe medication therapy is more important than diet and exercise, which may indicate their belief that taking drugs will compensate for an unhealthy diet and a lack of exercise, resulting in good medication adherence [30].

This study has some important strengths. The online survey allowed inclusion of patients who did not visit or stopped visiting hospitals and clinics compared to hospital or clinic-based surveys that do not include this category of patients. Therefore, it is possible that the results reflect their actual conditions. In addition, this study investigated the relationship between HLC, medication non-adherence, and lifestyle factors in a large sample of patients.

This study also has some limitations. First, the oldest participant in this study was in their 70 s, suggesting that the sample only included relatively young older adults. Therefore, the present results may not reflect the factors influencing medication non-adherence among older generations. The online mode of data collection may be a reason for this. Second, since no data were collected on the respondents’ income or educational history, socioeconomic factors could not be considered in depth. Moreover, since the instrument assessing trouble taking medication was developed specifically for this study, further assessment of the validity and reliability of the questionnaire is required to confirm the suitability of the questions used in this study. Finally, in this study, the survey period was short, and, additionally, the survey was conducted at the end of fiscal year in Japan (when many people are busy); therefore, the response rate to the questionnaire request email from the survey company was poor, resulting in relatively fewer responses. However, the overall number of respondents was larger than that calculated, and a sufficient number was secured for analysis.

The key to improving medication adherence is to educate patients via healthcare professionals and gain their trust to make them understand that results of drug treatment require the patients’ own efforts rather than relying on luck or supernatural factors. In addition, even relatively young patients may have problems with medication adherence, and understanding their practical problems may lead to solutions. Thus, improving medication adherence through healthcare professionals’ considerations of individual patients’ situations is based on the idea of shared decision making [31].

## 5. Conclusions

In our study, both intentional and unintentional medication non-adherence were observed among younger patients with lifestyle-related NCDs. Factors affecting medication non-adherence by patients with lifestyle-related diseases are uniquely related to their health awareness, lifestyle, and medication barriers, and differ by age group. Age-dependent personalized comprehensive care, including guidance on lifestyle and health related habits, is necessary to promote medication adherence.

## Figures and Tables

**Table 1 pharmacy-09-00090-t001:** Japanese version of the Health Locus of Control (HLC).

	Item	Dimension *
1	To stay healthy, it is good to pray hard and respect one’s ancestors.	S
2	I am healthy because I am lucky.	C
3	I take responsibility for my health.	I
4	Things that affect your health usually happen by chance.	C
5	I got sick because an undesired ghost possessed me.	S
6	Whether or not I recover from an illness depends on the warm support of my friends and family.	F
7	To stay healthy, I must pay attention to myself.	I
8	I got sick due to an ancestor’s undesirable fate.	S
9	Even if I feel sick, I only need a doctor.	Pr
10	Whether or not my illness improves will depend on my attitude.	I
11	If I donate to the gods and Buddha and ask for my safety, they will protect me from illnesses.	S
12	How long until my illness gets better depends on my luck.	C
13	It is thanks to God that I can stay healthy.	S
14	The improvement in my illness depends on my physician.	Pr
15	When I get sick, compassion from my family and others leads to recovery.	F
16	Whether or not I recover from a disease depends on my own efforts.	I
17	Whether or not the disease gets better depends on fate.	C
18	I am healthy because of my family’s compassion.	F
19	The improvement in the disease depends on the doctor’s judgment.	Pr
20	Whether or not I recover from my illness depends on whether there is someone who can help me.	F
21	How well I feel depends on my doctor’s skill.	Pr
22	It is a coincidence that I got sick.	C
23	Thanks to medical progress, I am staying healthy.	Pr
24	It is up to me to stay healthy.	I
25	Whether or not I will recover from my illness depends on the cooperation of my family.	F

* Internal (I), Family (F), Professional (Pr), Chance (C), and Supernatural (S).

**Table 2 pharmacy-09-00090-t002:** Unintentional and intentional non-adherence.

Unintentional Non-Adherence	
	Item
1	Please tell us how often you can imagine yourself having a hard time doing what your doctor suggested you to do.
2	Please tell us how often you can imagine yourself being unable to do what was necessary to follow your doctor’s treatment plans.
3	Please tell us how often you can imagine yourself forgetting to take your medications.
4	Please tell us how often you can imagine yourself missing taking your medications because you were away from home or busy with other things.
**Intentional Non-Adherence**	
	Item
1	Please tell us how often you can imagine yourself missing taking your medications because you seemed to need less medicine.
2	Please tell us how often you can imagine yourself missing taking your medications because you didn’t believe in the treatment your doctor was recommending.
3	Please tell us how often you can imagine yourself missing taking your medications because you wanted to avoid side effects or felt like the drug was toxic or harmful.
4	Please tell us how often you can imagine yourself missing taking your medications because you wanted to try alternative therapies (e.g., herbalist, homeopathic, or acupuncture treatments).
5	Please tell us how often you can imagine yourself missing taking your medications because the medication was too expensive.

**Table 3 pharmacy-09-00090-t003:** Characteristics of patients with lifestyle-related non-communicable diseases (NCDs).

			20–59 Years (*n* = 246)	Over 60 Years (*n* = 353)	*p*
Age (years)		Mean (*SD*)	46.1(10.1)	66.5(3.9)	
Sex	Male	*N* (%)	151(61.4)	202(57.2)	0.313
	Female	*N* (%)	95(38.6)	151(42.8)	
Diagnosed with lifestyle-related NCDs	Hypertension	*N* (%)	130(52.8)	221(62.6)	0.018
	Diabetes	*N* (%)	75(30.5)	63(17.8)	<0.001
	Dyslipidemia	*N* (%)	99(40.2)	130(36.8)	0.442
	Gout	*N* (%)	62(25.2)	51(14.4)	0.001
	Heart disease	*N* (%)	52(21.1)	55(15.6)	0.084
	Cerebrovascular disease	*N* (%)	39(15.9)	29(8.2)	0.006
	Cancer	*N* (%)	52(21.1)	64(18.1)	0.401

Notes: chi-square test.

**Table 4 pharmacy-09-00090-t004:** Unintentional and intentional non-adherence, HLC, lifestyle, and medication issues by patients’ characteristics.

		Total (*n* = 599)	20–59 Years (*n* = 246)	Over 60 Years (*n* = 353)
**Adherence (a)**				
Unintentional non-adherence	Mean (*SD*)	15.7 (3.3)	13.9 (4.0)	16.0 (2.9)
	*p*		<0.001	
Intentional non-adherence	Mean (*SD*)	21.5 (3.9)	19.8 (5.5)	22.4 (3.3)
	*p*		<0.001	
**HLC (a)**				
Supernatural	Mean (*SD*)	13.1 (5.8)	13.5 (6.1)	11.8 (5.0)
	*p*		0.001	
Internal	Mean (*SD*)	21.0 (4.3)	20.7 (5.0)	22.0 (3.6)
	*p*		0.001	
Chance	Mean (*SD*)	16.7 (4.6)	16.5 (4.9)	15.6 (4.5)
	*p*		0.034	
Family	Mean (*SD*)	18.7 (4.2)	18.4 (4.8)	19.3 (3.6)
	*p*		0.012	
Professional	Mean (*SD*)	18.0 (4.0)	18.0 (4.7)	18.4 (3.6)
	*p*		0.269	
**Lifestyle (b)**				
Alcohol consumption (over three days a week)	*N* (%)	255 (42.6)	102 (41.5)	153 (43.3)
	*p*		0.675	
Current smoker	*N* (%)	130 (21.7)	70 (28.5)	60 (17.0)
	*p*		0.001	
I feel I do not sleep enough	*N* (%)	170 (28.4)	90 (36.6)	80 (22.7)
	*p*		<0.001	
I do not care about my eating habits	*N* (%)	169 (28.2)	89 (36.2)	80 (22.7)
	*p*		<0.001	
I do not exercise	*N* (%)	249 (41.6)	106 (43.1)	143 (40.5)
	*p*		0.556	
**Trouble taking medication (b)**				
I have experienced side effects	*N* (%)	173 (28.9)	91 (37.0)	82 (23.2)
	*p*		<0.001	
Sometimes the medicine is hard for me to swallow	*N* (%)	96 (16.0)	58 (23.6)	38 (10.8)
	*p*		<0.001	
Sometimes I cannot handle medicine tools *	*N* (%)	39 (6.5)	30 (12.2)	9 (2.5)
	*p*		<0.001	
Sometimes I cannot hear what a healthcare professional says	*N* (%)	48 (8.0)	28 (11.4)	20 (5.7)
	*p*		0.014	
Sometimes I feel the explanations from doctors and pharmacists are difficult to understand	*N* (%)	66 (11.0)	38 (15.4)	28 (7.9)
	*p*		0.005	
Sometimes I forget what my doctor or pharmacist told me	*N* (%)	59 (9.9)	43 (17.5)	16 (4.5)
	*p*		<0.001	
I find it difficult to organize my medicines	*N* (%)	82 (13.7)	44 (17.9)	38 (10.8)
	*p*		0.015	

Notes: (a) *t*-test, (b) chi-square test. * e.g., insulin self-injection device, inhaler device for respiratory medicines.

**Table 5 pharmacy-09-00090-t005:** Correlation between unintentional and intentional non-adherence and types of HLC by age group.

	Supernatural	Internal	Chance	Family	Professional
20–59 years (*n* = 246)					
Unintentional non-adherence	−0.083	0.274 **	−0.016	0.154 *	0.101
Intentional non-adherence	−0.191 **	0.396 **	−0.037	0.196 *	0.160 *
Over 60 years (*n* = 353)					
Unintentional non-adherence	−0.106 *	0.170 **	−0.164 **	0.01	0.037
Intentional non-adherence	−0.222 **	0.232 **	−0.172 **	0.03	0.051

Notes: Pearson’s correlation coefficient * *p* < 0.05, ** *p* < 0.01.

**Table 6 pharmacy-09-00090-t006:** Unintentional and intentional non-adherence, lifestyle habits, and problems with taking medication by age group (multiple regression analysis).

	Unintentional Non-Adherence				Intentional Non-Adherence			
	20–59 Years (*n* = 246)		Over 60 Years (*n* = 353)		20–59 Years (*n* = 246)		Over 60 Years (*n* = 353)	
	β	*p*	β	*p*	β	*p*	β	*p*
**Lifestyle**								
Alcohol consumption (over three days a week)	0.110	0.079	0.034	0.524	0.147	0.020	0.059	0.275
Current smoker	0.280	<0.001	0.068	0.212	0.172	0.007	0.064	0.242
I do not care about my eating habits	−0.114	0.074	0.009	0.870	−0.136	0.034	0.042	0.442
I feel I do not sleep enough	−0.011	0.859	0.164	0.002	−0.055	0.374	0.091	0.089
I do not exercise	−0.102	0.115	0.050	0.355	−0.151	0.020	−0.005	0.920
Adjusted R^2^ *	0.141		0.026		0.136		0.006	
**Trouble taking medication**								
I have experienced side effects	0.150	0.009	0.141	0.005	0.104	0.080	0.117	0.022
Sometimes the medicine is hard for me to swallow	0.183	0.003	0.134	0.017	0.002	0.972	0.103	0.069
Sometimes I cannot handle medicine tools	0.045	0.552	−0.050	0.363	0.120	0.130	0.040	0.477
Sometimes I cannot hear what a healthcare professional says	0.245	0.004	−0.014	0.780	0.336	<0.001	0.041	0.421
Sometimes I feel the explanations from doctors and pharmacists are difficult to understand	−0.001	0.983	0.101	0.046	0.080	0.255	−0.030	0.556
Sometimes I forget what my doctor or pharmacist told me	0.147	0.035	0.305	<0.001	0.009	0.896	0.338	<0.001
I find it difficult to organize my medicines	0.132	0.039	0.077	0.173	0.164	0.014	0.035	0.542
Adjusted R^2^ *	0.468		0.228		0.429		0.210	

Notes: multiple regression analysis.* R^2^: Coefficient of determination.

## Data Availability

For any inquiries about the survey, please contact the corresponding author. Due to the privacy of data, the research team cannot share the data publicly.

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
