# Peer review of "Factors Associated with Medication Non-Adherence among Patients with Lifestyle-Related Non-Communicable Diseases"

_pharmacy, 2021, doi:10.3390/pharmacy9020090_

Round 1

Reviewer 1 Report

Manuscript ID: pharmacy-1167096

Title: Factors associated with medication non-adherence among patients with lifestyle-related non-communicable diseases

Medication non-adherence is, unfortunately, fairly common. Understanding why patients do not take their medications and investigating the factors involved in making this decision is crucial for physicians and other health professionals to provide better health services. Therefore, the authors conducted a cross-sectional study to explore the association between medication non-adherence and its factors in patients with non-communicable diseases (NCDs). 

Since this type of study is very common in many other countries, why this study is needed to conduct in the study area did not reflect in the problem statement. Therefore, the authors need to rewrite the background of this manuscript. The author may include the prevalence of NCDs, distribution among the population based on age group and mortality rate that might be relevant to medication non-adherence. This data required from the study location to highlights the problems existing now needed to resolve to contain NCDs death due to medication non-adherence.

Resolving some other issues might improve the quality of this manuscript as follows:

  1. Age group distribution is too broad (20-59 years). It should be split into several groups according to normal immunity strength among different ages of people.
  2. The reason for choosing 30,000 people randomly is needed to mention in the method section.
  3. The response rate is significantly less (authors should write the response rate at the beginning of results). Does this sample size enough to give a conclusion? If yes, why? Please write it in the manuscript.
  4. The data presentation is good, and the discussion is also written nicely.
  5. In conclusion, the authors can write the key findings of this study.
  6. Providing some solution on how to contain or decrease the non-adherence rate based on their findings and experience would increase this manuscript's scientific quality.

Reviewer 2 Report

Executive Summary

The manuscript titled “Factors associated with medication non-adherence among patients with lifestyle-related non-communicable diseases” discussed very practical and valuable research regarding medication non-adherence. The authors used a survey method to collect data and used different analytical methods to find the correlation and prevalence. Overall, the manuscript is scientifically written with good rationale and logical reasoning. Authors may perform minor revisions to further improve the quality and reader friendliness.

Major Comments

  • Line 77: please be consistent and describe the details regarding “current smoking status, sleep duration, eating habits, and exercise habits” in the same way as “alcohol consumption more than three days a week”. You may describe as “how many times per week/day”.
  • For all tables with p-value, please indicate the statistical method used, as mentioned in the methods section, in the footnote area. The goal is to improve reader friendliness.

Minor Comments

  • Table 3, since the t-test compares the means between two groups, please also provide the p-value for Sex-Female.
  • Section 4 Discussion:
    • “healthcare services should actively strive to raise patients’ internal HLC.” Based on Table 5, Supernatural factors also influence non-adherence. Therefore, it is also important for healthcare services to advocate the non-existence of supernatural power. If healthcare professionals are unable to do that, religious professionals may be involved. Please briefly discuss your thoughts about the S factor in HLC.
    • “It is important to follow these guidelines to address the dissatisfaction of patients aged 20–59, such as the inability to hear the healthcare professional’s explanations, and increase their trust.” Please also discuss what the pharmaceutical industry can do to help resolve this issue.
    • “Patients who follow a healthy diet often show good medication adherence”. As I mentioned above, a “healthy diet” or “healthy lifestyle” needs to be better defined with specific standards.
    • “the oldest participant in this study was 74 years of age, suggesting that the sample only included relatively young older adults.” A suggestion I have is to breakdown patients’ ages into finer groups. In most cases, clinical research breakdown age by 10 years interval.
